# Attenuation of *Pseudomonas aeruginosa* Virulence by Pomegranate Peel Extract

**DOI:** 10.3390/microorganisms10122500

**Published:** 2022-12-16

**Authors:** Samuele Peppoloni, Bruna Colombari, Davide Tagliazucchi, Alessandra Odorici, Cristiano Ventrucci, Aida Meto, Elisabetta Blasi

**Affiliations:** 1Laboratory of Microbiology and Virology, Department of Surgery, Medicine, Dentistry and Morphological Sciences with Interest in Transplant, Oncology and Regenerative Medicine, University of Modena and Reggio Emilia, Via G. Campi 287, 41125 Modena, Italy; 2Department of Life Sciences, University of Modena and Reggio Emilia, Via Amendola, 2—Pad. Besta, 42100 Reggio Emilia, Italy; 3Laboratory of Microbiology and Virology, School of Doctorate in Clinical and Experimental Medicine, University of Modena and Reggio Emilia, Via G. Campi 287, 41125 Modena, Italy; 4INCOS—Cosmeceutica Industriale, Funo di Argelato, 40050 Bologna, Italy; 5Department of Dentistry, Faculty of Dental Sciences, University of Aldent, 1007 Tirana, Albania

**Keywords:** anti-biofilm, *Pseudomonas aeruginosa*, pomegranate, virulence, autoinducers, phenolic compounds

## Abstract

*Pseudomonas aeruginosa* is an opportunistic pathogen often responsible for biofilm-associated infections. The high adhesion of bacterial cells onto biotic/abiotic surfaces is followed by production of an extracellular polysaccharidic matrix and formation of a sessile community (the biofilm) by the release of specific quorum-sensing molecules, named autoinducers (AI). When the concentrations of AI reach a threshold level, they induce the expression of many virulence genes, including those involved in biofilm formation, motility, pyoverdine and pyocyanin release. *P. aeruginosa* embedded into biofilm becomes resistant to both conventional drugs and the host’s immune response. Accordingly, biofilm-associated infections are a major clinical problem underlining the need for new antimicrobial therapies. In this study, we evaluated the effects of pomegranate peel extract (PomeGr) in vitro on *P. aeruginosa* growth and biofilm formation; moreover, the release of four AI was assessed. The phenolic profile of PomeGr, exposed or not to bacteria, was determined by high-performance liquid chromatography coupled to electrospray ionization mass spectrometry (HPLC-ESI-MS) analysis. We found that bacterial growth, biofilm production and AI release were impaired upon PomeGr treatment. In addition, the PomeGr phenolic content was also markedly hampered following incubation with bacterial cells. In particular, punicalagin, punicalin, pedunculagin, granatin, di-(HHDP-galloyl-hexoside) pentoside and their isomers were highly consumed. Overall, these results provide novel insights on the ability of PomeGr to attenuate *P. aeruginosa* virulence; moreover, the AI impairment and the observed consumption of specific phenolic compounds may offer new tools in designing innovative therapeutic approaches against bacterial infections.

## 1. Introduction

As an opportunistic pathogen, *Pseudomonas aeruginosa (P. aeruginosa)* causes severe infections in susceptible individuals, such as patients with acquired immunodeficiency syndrome (AIDS), cancer, severe burns or indwelling devices [1]. *P. aeruginosa* expresses numerous virulence factors that, through sophisticated regulatory mechanisms, allow it to adapt easily to many hostile environments [2]. Because of such adaptability and given the increasing drug-resistance, conventional antibacterial agents show limited efficacy against Pseudomonas [3]. As with many other microorganisms, besides those living in a planktonic form, *P. aeruginosa* is capable, as a survival strategy, of forming biofilm on medical devices and mucosal surfaces [4]. Accordingly, most hospital-acquired infections are associated with biofilm formation onto catheters, ventilator tubes, implants and medical prosthetic devices [5]. Therefore, early diagnosis of *P. aeruginosa* infection is crucial to promptly counteract the pathogen and, in turn, decrease mortality.

Microbial biofilm is a complex community of microorganisms’ adherent to living on abiotic surfaces and tightly embedded in a self-produced matrix (extracellular polymeric substances or EPS), primarily composed of polysaccharides, extracellular DNA (eDNA), proteins and lipids [6]. Importantly, the matrix, responsible for over 90% of the biofilm biomass, acts as a scaffold for adhesion to surfaces and, more importantly, protects sessile bacteria during stressful environmental conditions [7]. Indeed, when enclosed within the EPS, *P. aeruginosa* is protected from host immune responses and becomes less susceptible to antibiotics, compared to the planktonic counterpart [8]. Moreover, unlike planktonic cells, bacterial communities structured as biofilms exhibit an altered phenotype in terms of growth rate, expression of virulence factors and cell-to-cell communication systems [9].

For many microorganisms, the ability to form biofilm and finely modulate virulence factors expression is controlled by intercellular communication mechanisms, named quorum sensing system (QS), that function in a hierarchical manner, by means of signaling molecules and receptors, and are closely related to cell density [9]. In *P. aeruginosa,* four main QS systems have been identified, namely LasI/LasR, RhlI/RhlR [10], PqsABCDE/PqsR [11] and AmbBCDE/IqsR [12]. For each of these QS systems, *P. aeruginosa* synthesizes specific signal molecules, called autoinducers (AI): 3-oxododecanoyl-L-homoserine lactone (3-oxo-C12-HSL); the N-butanoyl homoserine lactone (C4-HSL); the 2- heptyl-3-hydroxy-4-quinolone (Pseudomonas quinolone signal–PQS) and the 2-(2-hydroxyphenyl)thiazole-4-carbaldehyde (integrated quorum detection signal–IQS) [13]. These AI are involved in the regulation of many genes related to motility, biofilm formation, host immune evasion, iron scavenging and antibiotic resistance [14]. For example, the LasI/LasR system reduces the polysaccharide matrix, thus inducing biofilm dispersion; it is also involved in the resistance to detergents [15]. The RhlI/RhlR system regulates the expression of rhamnolipids, which are important for the formation of a mature biofilm, its subsequent dispersion and resistance to host immune responses as well [16]. The PqsABCDE/PqsR system is mainly involved in biofilm formation and eDNA release, an important step for the stability of biofilm structure [17]. Finally, the fourth QS system, the IQS, influences the early stages of biofilm formation by affecting the swarming motility of *P. aeruginosa* [18]; in addition, IQS regulates the synthesis of siderophores, pyoverdine and pyochelin [19] and is responsible for the antibiotic resistance observed in the sessile bacteria within the biofilm [20]. Since antibiotic treatment has often limited efficacy against biofilm [21], a recent study has been focused on the regulatory mechanisms involved in QS with the aim of developing alternative therapies against *Pseudomonas* infections [22,23]. New molecules capable of interfering with QS signaling mechanisms may represent a successful antimicrobial new strategy. In this context, plant extracts and their derivatives, rich in polyphenols such as *Punica granatum L.*, could represent a new class of antimicrobial agents, capable of counteracting Pseudomonas’ virulence. Pomegranate peel extract (PomeGr) is an excellent source of biocompounds with biological activities and therapeutic properties, such as anti-inflammatory, antioxidant, antitumor and antimicrobial activities [24,25,26]. Recently, several studies have evaluated the antimicrobial activity of *Punica granatum* extract alone or in association with antibiotics. Interestingly, Abu El-Wafa WM et al. [27] have demonstrated that the combinations of pomegranate and rosemary extracts with antibiotics (namely piperacillin, ceftazidime, imipenem, gentamycin and levofloxacin) have a synergistic anti-Pseudomonas activity.

Here, we evaluate the effects of PomeGr on *P. aeruginosa* growth, biofilm formation and release of the AI, as well-known parameters linked to its pathogenicity. Moreover, the *P. aeruginosa* consumption of specific phenolic compounds present in PomeGr was also investigated.

## 2. Materials and Methods

### 2.1. Microbial Cells and Growth Conditions

The bioluminescent *P. aeruginosa* (BLI-*Pseudomonas*) strain P1242, expressing the luciferase gene and luciferin substrate under the control of a constitutive P1 integron promoter, was engineered by Choi and Schweizer [28]. This strain was maintained as previously described [29]. The bioluminescent signal was measured by a Fluoroskan reader (Thermo Fischer Scientific, Waltham, MA, USA) and expressed as relative luminescence units (RLU) as a parameter correlating with the number of viable cells [29]. Bacteria from −80 °C glycerol stocks were seeded onto tryptic soy agar (TSA) (OXOID, Milan, Italy) plates and incubated overnight at 37 °C. The resulting colonies were then collected, added to 10 mL of tryptic soy broth (TSB) (OXOID, Milan, Italy) and further incubated overnight at 37 °C by gentle shaking. The day after, the OD_595_ of the bacterial suspension was spectrophotometrically measured (by TECAN Sunrise, Männedorf, Switzerland) and the obtained value was converted into colony-forming units (CFU)/mL, according to an internal standard curve. Then, the bacterial suspension was appropriately diluted to obtain the final working condition of 10^6^ bacteria/mL.

### 2.2. Pomegranate Peel Extract

The *Punica granatum* L. peel extract (PomeGr), supplied by INCOS COSMECEUTICA INDUSTRIALE (Bologna, Italy) and produced by PHENBIOX SRL (Bologna, Italy), contained the peel extract (22.5% *w*/*w*), saccharomyces ferment lysate filtrate, citric acid, sodium benzoate and potassium sorbate, as detailed elsewhere [26]. The same solution without PomeGr was used as a negative control (neg-C). The PomeGr and the neg-C were filtered through a 0.22 µm membrane and stored at 4 °C until tested in the assays.

### 2.3. Total Microbial Growth, Biofilm Formation and Regrowth Assays

The PomeGr was serially diluted and plated (100 μL/well) in black 96-well microtiter plates with transparent bottoms (PerkinElmer, Milano, Italy). The bacterial suspension (10^6^ cells/mL in TSB, prepared as detailed above) was added to the plates (100 μL/well) and incubated at 37 °C for 24 h. The fluorescent signal was hourly measured; the RLU values represented the amounts of live cells in the treated and untreated groups. After treatment, the samples were washed two times and the bioluminescent (BLI) signal was measured to quantify the 24 h-old biofilm, as previously detailed [30,31]. Next, to evaluate the PomeGr inhibitory effects on Pseudomonas regrowth, fresh medium was added to each well and the plate was further incubated at 37 °C; microbial growth was kinetically checked (24 to 48 h) by measuring the RLU every hour.

### 2.4. Mass Spectrometry Analysis of PomeGr Extract

The phenolic profile of PomeGr extract (diluted 1:16), exposed or not to *P. aeruginosa* (10^6^ cells/mL) for 24 h at 37 °C, was assessed by high-resolution mass spectrometry (HR-MS). The cell-free supernatants were ultrafiltered with Amicon-Ultra 0.5 tubes (14,000 rpm; 15 min), diluted 1:1 in a methanol/water solution (5:95 *v*/*v*) and injected in an Ultimate 3000 UHPLC system coupled to a Q Exactive Mass Spectrometer (Thermo Scientific, San Jose, CA, USA). Chromatographic separation was performed by a C18 column (Hypersil Gold C18, 100 × 2.1 mm, 1.9 μm particle size; Thermo Scientific, San Jose, CA, USA). The mobile phases consisted of (A) water acidified with formic acid (0.1%) and (B) acetonitrile acidified with formic acid (0.1%). The flow rate was set at 0.5 mL/min. The elution gradient began at 2% B and after 0.5 min started to linearly increase to 18% B in 27 min and afterwards to 98% B in 4.5 min. In order to wash the column, solvent B was maintained at 98% for 3 min before returning to the initial conditions. The MS and MS/MS parameters are fully reported in Colombari et al. [26]. The relative amount of the identified phenolic compounds was determined by integrating the area under the peak (AUP) from the extracted ion chromatograms (tolerance ± 5 ppm).

### 2.5. Mass Spectrometry Analysis of P. aeruginosa AI

Cell-free supernatants from 24 h-old *P. aeruginosa* cultures (10^6^/mL), treated or not with PomeGr (1:16 dilution), were harvested, filtered and analyzed by HPLC-ESI-MS, as previously detailed [31]. The relative amount of the four AI was determined by integrating the area under the peak (AUP) from treated and untreated cultures (tolerance ± 5 ppm).

### 2.6. Statistical Analysis

Statistical analysis was performed by using GraphPad Prism 8 software. Statistical differences between groups were evaluated with the one-way ANOVA followed by Sidak’s multiple comparisons tests, as previously detailed [26]. Values of *p* ≤ 0.001 were considered statistically highly significant.

## 3. Results

### 3.1. PomeGr Effects on Pseudomonas Growth and Biofilm Formation

Initially, we assessed in real time the effects of PomeGr on *P. aeruginosa* planktonic cell growth by using a previously established BLI-based assay [30]. As depicted in Figure 1A–C, the curves of the untreated samples (medium) showed the expected time-dependent increase in RLU that reached their maximal values after 8 h.

Similarly, when exposed to the neg-C, *P. aeruginosa* showed curves comparable to those of the medium, independent of the dilution tested. In contrast, at the 1:8 dilution, PomeGr significantly affected the RLU already at time 0 (drop of 2 Logs); interestingly, the curve consistently remained below the controls at all the timepoints tested, with a 3-Log difference detected at 24 h. At the 1:16 PomeGr dilution, the inhibition observed was partial and gradually disappeared in that the RLU reached the values of the controls (medium and neg-C) between 17 and 24 h. Furthermore, no effects were found using the 1:32 dilution. Next, to investigate whether PomeGr would affect *P. aeruginosa* biofilm formation, after 24 h of treatment, the wells were washed to remove non-adherent bacteria and the residual bioluminescent signal was measured. As shown in Figure 1D, PomeGr significantly affected biofilm production in a dose-dependent manner. In particular, >99% inhibition was observed when employing the 1:8 dilution, while about 55% and 33% impairment were detected at 1:16 and 1:32 dilutions, respectively. When assessed against neg-C, the PomeGr significantly affected biofilm production at all the dilutions tested (*p* < 0.001); furthermore, at 1:64 and 1:128, the impairments were still significant (*p* < 0.01; data not shown).

In order to evaluate whether the PomeGr inhibitory effects on Pseudomonas biofilm persisted over time, we next focused on the 24–48 h timeframe. Accordingly, after 24 h incubation, PomeGr was removed from the cells and fresh medium was added to each well. The plate was again incubated at 37 °C and microbial load was further kinetically assessed (from 24 to 48 h) by measuring the RLU. As shown in Appendix A (Appendix A), cells exposed to the 1:8 dilution slowly regrew, reaching the control/neg-C treated cells at about time 38 h, while the 1:16 and 1:32 dilutions were ineffective.

Since the 1:16 PomeGr dilution significantly impaired Pseudomonas biofilm production without affecting total growth, this condition was chosen for further analyses.

### 3.2. Phenolic Compounds Profile of PomeGr Exposed or Not to P. aeruginosa

Pomegranate peel extract is known to be a rich source of phenolic compounds, including flavonoids (anthocyanins, catechins and other complex flavonoids) and hydrolysable tannins (punicalin, pedunculagin, punicalagin, gallic acid and ellagic acid esters of glucose), previously reported as exerting numerous biological activities such as antioxidant, antimicrobial, anti-proliferative and anti-inflammatory activities [32,33,34,35].

By comparing the PomeGr MS and MS/MS profiles with those shown by the literature, several compounds were identified and the MS details on the phenolic compounds profile of PomeGr were given in our recently published manuscript [26]. Appendix A shows the data about the relative quantification of each phenolic compound (expressed as AUP) detected in the PomeGr, that had been exposed or not to *P. aeruginosa.* Furthermore, the relative percentage decrease was calculated. Figure 2 details the overlapping chromatographic peaks of five compounds that displayed the highest decrease after PomeGr incubation with *P. aeruginosa* cells. In particular, following PomeGr exposure to *P. aeruginosa* (grey lines) and, in comparison, with the control (PomeGr alone; black lines), a significant decrease in pedunculagin (decrease ranging from 58.7 to 88%), punicalagin (decrease ranging from 32 to 72%), granatin (decrease ranging from 94.8 to 99.6%), punicalin (21.1% decrease) and di-(HHDP-galloyl-hexoside)-pentoside (82.4% decrease) was observed.

### 3.3. PomeGr Effects on AI Release by P. aeruginosa

*P. aeruginosa* produces several AI signal molecules involved in Pseudomonas biofilm formation and bacterial virulence as well. By HPLC-ESI-MS analysis, we measured the AI production by *P. aeruginosa* exposed or not to 1:8 and 1:16 dilutions of PomeGr. Using cell-free supernatants, we obtained several chromatograms, through which the AUP of the various elution peaks were measured and the semi-quantitative assessments of the specific products were performed. Table 1 shows the peak areas of four AI, namely 3-oxo-C12-HSL, C4-HSL, PQS and IQS, detected in the supernatants of cells exposed or not to PomeGr. A marked reduction in the Al content was observed following PomeGr treatment; the observed effect was dose-dependent and ranged between 0.5 and 4 Log; moreover, the amount of IQS also dropped to undetectable levels.

## 4. Discussion

Here, we show that PomeGr affects *P. aeruginosa* growth, biofilm formation and AI release, implying a marked impairment of its virulence. In addition, the consumption of specific phenolic compounds by *P. aeruginosa* suggests their direct involvement in the antibacterial activity.

*P. aeruginosa* is an opportunistic pathogen frequently involved in biofilm-related infections that are difficult to treat with conventional drugs. Several studies have been performed on the antimicrobial activity of *Punica granatum* L., with the aim of successfully countering infections by using alternative approaches [25,27,32,34,35]. Here, we demonstrate the ability of PomeGr to significantly impair *P. aeruginosa* growth and biofilm formation in a dose-dependent manner. Indeed, as shown in Figure 1, the kinetically evaluated microbial growth remains lower than that observed in medium or neg-C treated cells, at PomeGr dilutions 1:8 or below (data not shown). At 1:16 PomeGr dilution, the RLUs progressively increase, reaching the control levels between 16–18 h, while at 1:32 dilution no effects have been detected. Furthermore, as depicted in Figure 1D, PomeGr significantly affects biofilm formation in a dose-dependent manner. In particular, more than 99% inhibition occurs at 1:8 (Figure 1D) or less (data not shown) while, at 1:16 and 1:32 dilutions, the decrease ranges from 54% to 24%, respectively. In our experiments, the neg-C has no effects on *Pseudomonas*, implying that the additives/preservatives contained in the extraction solution have no role or toxic effects by themselves. In any case, we have used concentrations of potassium sorbate, sodium benzoate and citric acid below the MIC previously established by several studies against *P. aeruginosa* [36,37].

In our model, the anti-Pseudomonas effect is reversible, since the removal of PomeGr allows bacterial regrowth (see the Appendix A). Whether the PomeGr-treated population may exhibit different phenotypes, i.e., susceptible, resistant and/or tolerant cells, as proposed by other studies obtained exposing Pseudomonas to antibiotics [38,39], remains to be investigated. Interestingly, initial evidence exists on the synergism between pomegranate extract and conventional antibiotics against Pseudomonas [27], further emphasizing the potential relevance of such a natural product in the design of novel antimicrobial protocols. Here, we show that the best anti-Pseudomonas effects occur at 1:8 and 1:16 dilutions, the same conditions that had also been found to exert the highest effects against *Candida albicans* [26]. Taken together, our previous [26] and present data suggest that the same molecules act against bacterial and fungal target cells as well.

*Punica granatum* L. has been described as an excellent source of biocompounds, including phenolic acids, flavonoids and hydrolyzable tannins, mainly ellagitannins and gallotannins, each of them with beneficial properties on human health [35,40,41]. In line with the literature [41], we recently found that the PomeGr, used also in the present study, is mainly composed by ellagitannins such as pedunculagin and its isomers, ellagic acid-hexoside, punicalagin and its isomers, punicalin, granatin and di-(HHDP-galloyl-hexoside)-pentoside and its isomers [26]. Here, we demonstrated a remarkable consumption of polyphenols by PomeGr exposure to *P. aeruginosa*. As shown in Appendix A and in Figure 2, a strong reduction in the levels of pedunculagin (decrease ranging from 58.7 to 88%), punicalagin (decrease ranging from 32 to 72%), granatin (decrease ranging from 94.8 to 99.6%), punicalin (decrease of 21.1%) and di-(HHDP-galloyl-hexoside)-pentoside (decrease of 82.4%) occurs, highlighting the involvement of these molecules in the anti-Pseudomonas activity. It should be noted that a similar trend of consumption had been observed exposing PomeGr to fungal cells [26]. This further emphasizes the likelihood that the same effector molecules act against bacterial and fungal cells as well. It is worth pointing out that the compounds showing the highest reduction (pedunculagin, punicalagin, granatin and di-(HHDP-galloyl-hexoside)-pentoside) contain in their structures at least one hexahydroxydiphenoyl moiety, suggesting its pivotal role in the antimicrobial and antifungal activities exerted by PomeGr.

*P. aeruginosa* biofilm formation is finely regulated by cell-to-cell communication systems that function in a hierarchical manner, by means of signaling molecules and receptors [10]. In particular, four main QS systems have been identified: LasI/LasR, RhlI/RhlR [11]; PqsABCDE/PqsR [12] and AmbBCDE/IqsR [13]; each of them synthesizes its own specific signal molecules, namely 3-oxo-C12-HSL, C4-HSL, PQS and IQS, respectively. Here, we show that all of the four AI are constitutively produced Pseudomonas, being detected in 24 h-old supernatants (Table 1). Interestingly, a marked reduction in the Al content is observed upon PomeGr treatment; particularly, a drop, ranging from 0.5 to 4 Log, occurs for 3-oxo-C12-HSL, C4-HSL and PQS, while the amounts of IQS drop to undetectable levels. These data provide the first evidence that PomeGr impairs *P. aeruginosa* AI production, although to a different extent, depending on the molecule considered. Initial literature [41] describes a close relationship between the flavonoid structure of certain compounds and some biological effects, such as impairment of pyocyanin and elastase production by Pseudomonas. In particular, Paczkowski et al. [41] have demonstrated that flavonoids are direct inhibitors of the QS receptor, LasR; yet, they do not function via a competitive mechanism involving displacement of their natural ligand, the AI 3-oxo-C12-HSL; rather, they prevent binding to the respective promoter region, thereby inhibiting the expression of downstream genes. Furthermore, it has been shown [42] that a methanolic extract of *Terminalia chebula,* containing ellagic acid-derivatives and flavogallonic acid-derivatives (structurally related to the hexahydroxydiphenoyl moieties), is able to inhibit *P. aeruginosa* biofilm formation, via downregulation of the *lasIR* and *rhlIR* gene expression, in turn resulting in a decreased production of AI [42]. The molecular mechanisms involved in the above-described inhibition remain to be investigated; however, the hierarchical organization of the various QS systems, controlled by the LasI/LasR, may explain the impairment of the four AI, observed in our model. Given the crucial role of such signal molecules that, through their specificity and concentration, finely regulate not only biofilm formation but also the expression of genes involved in other processes, such as stress-tolerance and host–microbe interaction [39], we may conclude that bacterial virulence is indeed profoundly attenuated upon PomeGr treatment.

Despite the limitations intrinsic to any in vitro model, these results add new insights on the molecular mechanisms occurring during Pseudomonas biofilm development and its impairment by PomeGr. It remains to be established whether other microorganisms and/or multispecies biofilm may be successfully treated with PomeGr, hopefully by targeting other QS systems.

## 5. Conclusions

The results of this study emphasize the role of PomeGr as a natural antimicrobial compound against *P. aeruginosa.* Indeed, upon exposure to PomeGr, (i) bacterial growth and (ii) biofilm formation are affected; simultaneously, (iii) the consumption of specific phenolic compounds and (iv) the impairment of AI production by *P. aeruginosa* are detected. Because the AI-regulated biofilm production is strictly related to bacterial virulence, we may consider PomeGr as an interesting alternative to conventional drugs for the design of novel therapeutic protocols focusing on virulence attenuation of *P. aeruginosa*.

## Figures and Tables

**Figure 1 microorganisms-10-02500-f001:**
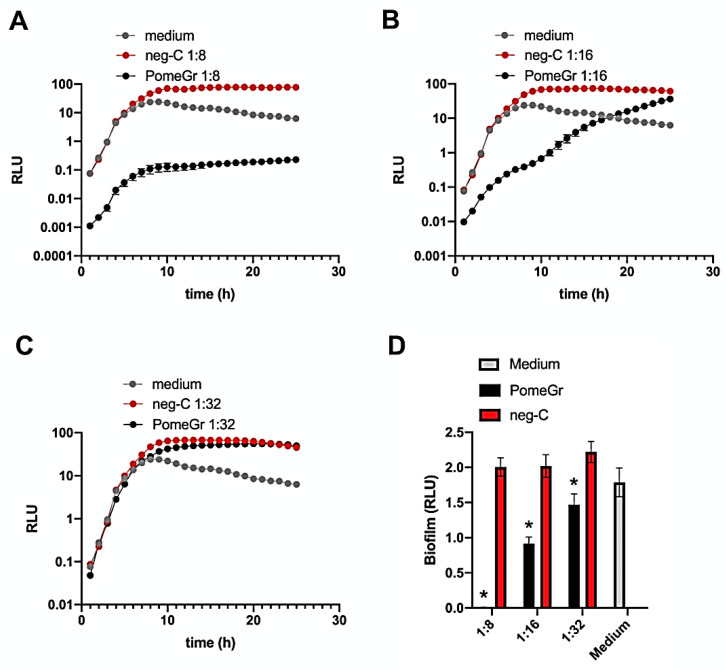
The PomeGr effects on *P. aeruginosa* growth and biofilm formation. *P. aeruginosa* (10^6^/mL, 100 µL/well) was exposed for 24 h to PomeGr or neg-C (100 µL/well) at dilutions of 1:8, 1:16 and 1:32. By BLI assay, the total microbial load was kinetically measured every hour (Panel **A**: 1:8 dilution; Panel **B**: 1:16 dilution; Panel **C**: 1:32 dilution). The samples were then washed and the biofilm formation was quantified by an additional bioluminescence reading (Panel **D**). The results shown are the mean (±SEM) of the RLU from 9–12 replicates of three independent experiments. The asterisk indicates *p* ≤ 0.001 (PomeGr vs. neg-C).

**Figure 2 microorganisms-10-02500-f002:**
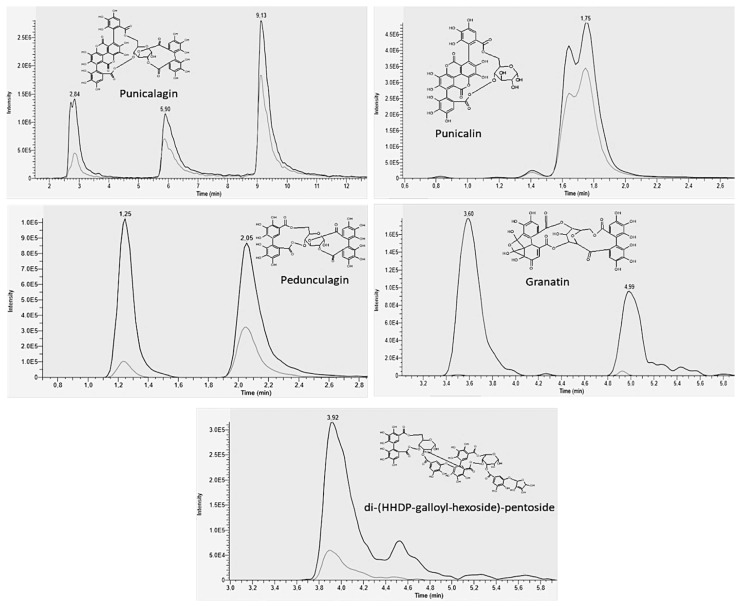
Phenolic compounds decrease upon PomeGr exposure to *P. aeruginosa*. Overlapped peaks of 5 phenolic compounds in PomeGr extract, assessed alone (black line) or upon exposure to *P. aeruginosa* (grey line). The data were obtained by a pool of 4 replicates from a representative experiment. Each panel also shows the chemical structure of each compound.

**Table 1 microorganisms-10-02500-t001:** AI release by *P. aeruginosa* exposed or not to PomeGr.

Treatment	3-oxo-C12-HSL	C4-HSL	PQS	IQS
Medium	1.65 × 10^8^	3.15 × 10^8^	2.51 × 10^10^	2.09 × 10^6^
PomeGr 1:8	1.89 × 10^6^	3.99 × 10^7^	4.42 × 10^6^	n.d.
PomeGr 1:16	6.58 × 10^7^	3.92 × 10^7^	1.93 × 10^8^	n.d.

The AI were measured on cell-free supernatants from *P. aeruginosa*, treated or not with PomeGr, by HPLC-MS. By chromatogram analysis, the different elution peaks were identified and their areas used for semiquantitative evaluation of each molecule. The results shown are from a pool of 4 replicates of a representative experiment. n.d. not detectable.

## Data Availability

The data presented in this study are available on request from the corresponding authors.

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
