# Peer review of "Attenuation of Pseudomonas aeruginosa Virulence by Pomegranate Peel Extract"

_microorganisms, 2022, doi:10.3390/microorganisms10122500_

Round 1

Author Response

Referee 1

1th comment: - The abstract needs to be developed further and structured into background, case presentation and conclusion.

Answer: The abstract was in part expanded by adding some details on the background (concerning the autoinducers), while the general  format has not been modified to respect  the style of  the journal.

2nd comment: - For biofilm assay, the authors reported that dilution at 1:16 and 1:32 if effective, what’s the OD of bacterial growth you have started? Please mention it in the section of materials and methods

Answer: As suggested by the Referee, the Materials and Methods section (paragraph 2.1, rows 143-147) now provides details on how the working conditions were achieved (initial OD595 reading, conversion in CFU/ml and subsequent appropriate dilution); please, consider that the working suspension (106/ml) is not spectrophotometrically readable.

Reviewer 2 Report

Dear authors,

This manuscript describes a relevant study regarding the ability of pomegranate extract to counteract the biofilm formation and quorum sensing induction in Pseudomonas aeruginosa. The manuscript is well written, apart from some minor mistakes described below. The authors must check all references as some numbers do not seem to correlate between the text and the reference section.

Best regards,

42. Change Fragile for susceptible

129 Change quantify for quantity

162 Change PomeGR for Pomegr

264 The sentence “the effect is dose-dependent” , should be modified to facilitate understanding (e.g.: in a dose-dependent manner)

265 Uncapitalize Medium

283 remove “possibly” after the word that

All reference numbers should be checked as for example, Abu El-Wafa WM et al. [28] (line 95) does not match with the reference number found in the reference section (27 in this case)

Author Response

Referee 2

The manuscript is well written, apart from some minor mistakes described below. The authors must check all references as some numbers do not seem to correlate between the text and the reference section.

Answers: As required, the following changes were made.

  1. Change Fragile for susceptible: (now row 44) done
  2. Change quantify for quantity: (now row 162) done
  3. Change PomeGR for Pomegr: (now row 209) done
  4. The sentence “the effect is dose-dependent”, has been changed in.. we demonstrate the ability of PomeGr to significantly impair P. aeruginosa growth and biofilm formation “in a dose-dependent manner”. (now rows 321)
  5. Medium has been changed in “medium”: (now row 322) done 
  6. remove “possibly” after the word: (now row 340) done

All reference numbers should be checked.

Answers: We have checked all references in the test, as well as in the reference section, as recommended by the Referee.

Reviewer 3 Report

            REVIEW REPORT

The study by Samuele Peppoloni et al. "Attenuation of Pseudomonas aeruginosa Virulence by Pomegranate Peel Extract", sought to investigate whether pomegranate peel extract had any impact on P. aeruginosa growth, biofilm formation, and AI release-all well-known factors associated with the pathogenicity of the organism.

ABSTRACT & INTRODUCTION

Well done, the concept is correct but in some cases need to complete in the correct way. The suggestions are in bold:

·         Page 1 Line 42: Check the abbreviations throughout the manuscript and introduce the abbreviation when the full word appears the first time in the text and then use only the abbreviation e.g., acquired immunodeficiency syndrome (AIDS), EPS…

·         Page 2 Line 91: “Pomegranate peel extract (PomeGr) is an excellent source of biocompounds…” Insufficient information previously available in the literature regarding the active constituents of pomegranate peels.

·         Are pomegranate peels approved for human consumption? Does it not contain toxic compounds that may be harmful to health?

Material and Methods

·         “Microbial Cells and Growth Conditions The source of the strain used in this work is not mentioned.

·         Page 3 Line 119: “The same solution without PomeGr was used as negative control (neg-C)”. Why was there no positive control (a standard antibacterial agent) along with a negative control in this experiment, both at similar concentrations?

·         Page 3 Line 126: Please explain what this abbreviation "RLU" means; and what is the unit of RLU?

·         Page 3 Line 157: “Statistical differences between groups were evaluated with the one-way ANOVA followed by Sidak’s multiple comparisons tests.” [Choose a reference].

Results and discussion

Well done, the figures and tables are clear. No observations.

·         Page 5 Lines 208-212: This section should be moved to the introduction section.

Conclusions

·         The conclusions need to be rewritten in point-by-point form with key numbers obtained from the study.

REFERENCES

·         Please follow the guidelines of the journal about references

Author Response

Referee 3

ABSTRACT & INTRODUCTION 

Well done, the concept is correct, but in some cases need to complete in the correct way.

The suggestions are in bold:

-    Page 1 Line 42: Check the abbreviations throughout the manuscript and introduce the abbreviation when the full word appears the first time in the text and then use only the abbreviation e.g., acquired immunodeficiency syndrome (AIDS), EPS…

Answer: Done, as suggested.

-  Page 2 Line 91: “Pomegranate peel extract (PomeGr) is an excellent source of  biocompounds…” Insufficient information previously available in the literature regarding the active constituents of pomegranate peels.  

Answer: Several papers that appeared in literature describe the biological activity of pomegranate peel (see Reff.  24,25,26,27,34, 35)

-  Are pomegranate peels approved for human consumption?  

Answer: Yes, at least in Italy, the pomegranate peels are approved for human consumption, in particular,  as food supplementation (see “Gazzetta Ufficiale della Repubblica Italiana, July 21, 2012, n.169).

- Does it not contain toxic compounds that may be harmful to health?
 Answer: To our knowledge, there are no data on toxicity so far. In our hands, PomeGr did not exert toxic effects on human epithelial cells in vitro, as previously established by using a continuous cell line (ref 26)

Material and Methods 

  • “Microbial Cells and Growth ConditionsThe source of the strain used in this work is not mentioned. Answer:  We have now mentioned the source of the BLI Pseudomonas strain (see Materials  and Methods section, paragraph 2.1 rows 134-137)

 ·   Page 3 Line 119: “The same solution without PomeGr was used as negative control (neg-C)”. Why was there no positive control (a standard antibacterial agent) along with a negative control in this experiment, both at similar concentrations?
Answer: In a previous study (Ref. 29; Pericolini et al.), we analyzed in detail the effect of gentamicin  on P. aeruginosa biofilm formation (using that same strain). In that model, a 24 h treatment (4 ug/ml) caused more than 50% biofilm reduction. Here, being the focus on the anti-Pseudomonas efficacy of a natural compound, the pomegranate, we felt as non-appropriate a comparative analysis with conventional antibacterial drugs. In any case, in the discussion, we added a sentence that, by mentioning the initial evidence on the synergism between pomegranate and antibacterials (see ref. 27), underlines the potential relevance of such a natural product in the design of novel antimicrobial protocols (see in Discussion session, rows 340-343).   

     Page 3 Line 126:

  • Please explain what this abbreviation "RLU" means
    Answer: as mentioned in the rows 138,139 of the original paper, the acronym “RLU” means Relative Luminescence Units, a parameter known to correlate with the number of viable cells (see ref. 29);
  • and what is the unit of RLU?

Answer: the RLU are indeed units “relative” to the system/instrument used (we had used  the Fluoroskan microplate reader); thus, the RLU do not have any physical meaning but they are very useful parameter for quantitative analysis within a specific experimental model. 

   Page 3 Line 157: “Statistical differences between groups were evaluated with the one-way ANOVA followed by Sidak’s multiple comparisons tests.”[Choose a reference].

Answer: Reference 26 has been now added in the Materials and Methods section, paragraph 2.6 row 206

Results and discussion 

Well done, the figures and tables are clear. No observations.

  • Page 5 Lines 208-212: This section should be moved to the introduction section.

Answer: this part has actually been removed from the results, since it was already present in the discussion section (rows 328-332) of the revised manuscript)

Conclusions

  • The conclusions need to be rewritten in point-by-point form with key numbers obtained from the study

Answer: Done, accordingly to the Referee’s request (see last paragraph).  

REFERENCES

  • Please follow the guidelines of the journal about references

Answer: Done accordingly to the journal guidelines.